# Proteomic Studies of Psoriasis

**DOI:** 10.3390/biomedicines10030619

**Published:** 2022-03-07

**Authors:** Vladimir V. Sobolev, Anna G. Soboleva, Elena V. Denisova, Eva A. Pechatnikova, Eugenia Dvoryankova, Irina M. Korsunskaya, Alexandre Mezentsev

**Affiliations:** 1Centre of Theoretical Problems of Physico-Chemical Pharmacology, Russian Academy of Sciences, 119334 Moscow, Russia; annasobo@mail.ru (A.G.S.); evdenissova@rambler.ru (E.V.D.); edvoriankova@gmail.com (E.D.); marykor@bk.ru (I.M.K.); 2Scientific Research Institute of Human Morphology, 117418 Moscow, Russia; 3Moscow Scientific and Practical Center of Dermatovenereology and Cosmetology, 119071 Moscow, Russia; 4Department of Dermatology and Cosmetology, Pirogov Russian National Research Medical University, 117997 Moscow, Russia; evavasilevska@gmail.com

**Keywords:** mass spectrometry, LC-MS/MS, SOMAscan™, proximity extension assay, psoriasis, comorbidities, biomarkers, predisposition, risk factors

## Abstract

In this review paper, we discuss the contribution of proteomic studies to the discovery of disease-specific biomarkers to monitor the disease and evaluate available treatment options for psoriasis. Psoriasis is one of the most prevalent skin disorders driven by a Th17-specific immune response. Although potential patients have a genetic predisposition to psoriasis, the etiology of the disease remains unknown. During the last two decades, proteomics became deeply integrated with psoriatic research. The data obtained in proteomic studies facilitated the discovery of novel mechanisms and the verification of many experimental hypotheses of the disease pathogenesis. The detailed data analysis revealed multiple differentially expressed proteins and significant changes in proteome associated with the disease and drug efficacy. In this respect, there is a need for proteomic studies to characterize the role of the disease-specific biomarkers in the pathogenesis of psoriasis, develop clinical applications to choose the most efficient treatment options and monitor the therapeutic response.

## 1. Introduction

Psoriasis is one of the most prevalent chronic skin disorders. The common symptoms of psoriasis include red patches covered with flaking silvery scales, itching, burning, or soreness, and thickened, pitted, or ridged nails [1]. The histological analysis of psoriatic skin reveals epidermal hyperplasia and parakeratosis. The microcapillaries of the upper dermis dilate. In addition, the skin becomes infiltrated by immune cells predominantly, lymphocytes, monocytes, and neutrophils [2]. In line with these findings, psoriatic keratinocytes exhibit hyperproliferation and altered terminal differentiation [3].

Psoriasis is a multigenic and multifactorial disorder. The heritability of the disease is complex due to the contribution of multiple susceptibility loci [4]. The genome-wide association studies (GWAS) reveal that the progression of psoriasis changes the expression of several thousand genes. However, these changes explain less than 25% of the disease heritability [5]. Moreover, there is a need for further research of disease biomarkers suitable to monitor the therapeutic response, assess the treatment efficacy, and drug-related toxicity. In this respect, rapidly evolving proteomic methodology frequently used for the systemic analysis of disease-associated proteins, their identities and quantities can be of great help to accomplish these tasks [6].

The term “proteomics” proposed by Mark Wilkins in 1994 applies to the studies that describe and compare sets of proteins expressed in cultured cells, tissues, or the entire organism in desired conditions [7]. Different experimental techniques (gel-based and gel-free high throughput screening techniques, ICAT, SILAC, iTRAQ, etc.) are available. They allow localizing proteins of interest, identifying their isoforms, posttranslational modifications, and interactions [8]. The proteomic analysis also provides the opportunity to explore the molecular mechanisms contributing to the pathogenesis of diseases [9]. The previous and recent proteomics studies of psoriasis are briefly summarized in Table 1 and discussed as follows.

## 2. The Analysis of Differentially Expressed Proteins in the Skin

For their studies, the authors of the earliest papers used 2D gel electrophoresis followed by mass spectrometry, MS (Figure 1). In 2005, Carlén et al. [10] identified 11 differentially expressed proteins (DEPs) in lesional skin: nine proteins with increased and two proteins with decreased expression, compared to normal-looking skin of psoriasis patients. We note that some protein names used in their paper (e.g., “SCCA2, low pI” and “SCCA2, high pI”) would look very unusual for modern scientists. Moreover, Carlén et al. did not use UniProt IDs as UniProt was found only three years earlier [52]. At the same time, the authors demonstrated significant differences in proteomic profiles between skin lesions of guttate and plaque psoriasis. They also concluded that longer-lasting disease produces greater changes in skin proteome.

Ryu et al. [14] identified 35 upregulated proteins comparing lesional and normal-looking skin of psoriasis patients. Then, the authors verified their expression by independent methods, namely Western blot and immunohistochemistry. The authors, probably, were among the first to perform ontology analysis on the set of identified proteins. Respectively, they suggested that 11 identified proteins, namely GSTP1, HSPB1, HSP90B1, HSPA5, PRDX2, YWHAZ, PDIA3, YWHAE, SFN, TUBB2C, and YWHAB participated in the regulation of apoptosis. In turn, eight others, namely S100A7, TUBB2C, S100A9, APCS, PRDX2, SERPINA1, TF, and YWHAZ contributed to the “defense response” of the disease-affected cells. Moreover, they reported that the last six proteins contributed to the inflammatory response. In addition, they linked a higher expression of GSPT1 and PRDX2 to the prevention of cells damages caused by reactive oxygen species (ROS) [53]. Looking through the results of their analysis, we must keep in mind that ten years ago, there was little known about the functions of the individual proteins that Ryu et al. identified in their study. Respectively, the direct implementation of their results elsewhere may be confusing. However, their main accomplishment was that they grouped disease-associated DEPs with the same functions and proposed that these proteins acted as groups in the diseased tissue.

Using a similar technique, Piruzian et al. [15] reported ten proteins differentially expressed in lesional skin, namely KRT16, SERPINB4, SERPINB3, S100A9, S100A7, KRT17, ENO1 KRT14, ENO1, and LGALS7B compared to normal-looking skin of the same patients. Upregulation of these proteins was associated with the induction of their coding genes. In this regard, we would like to acknowledge a limited capacity of gel-based proteomics to separate acidic, basic, hydrophobic, and low abundance proteins. These days, this and similar studies have limited practical value as their authors preferred to identify the most abundant proteins instead of using less expensive and faster non-MS techniques for the same proposes.

Several extensive, large-scale comparative proteomic studies were aimed to identify and characterize the molecular mechanisms of psoriasis. The paper by Schonthaler et al. [16] provided important insights into the role of S100A8-S100A9 and complement C_3_ in the pathogenesis of the disease. Presumably, the authors were the first who analyzed the whole proteome of the human lesional epidermis and reported ten of the most upregulated proteins, such as S100A8 and S100A9. Then, using *JunB*-*Jun* double knockout mice, they also characterized the role of S100A8-S100A9 dimer in the inflammatory response triggered by the disease.

To reduce the complexity of experimental data, Williamson et al. [18] replaced the analysis of total protein with secretomes. The authors obtained and cultured dermal cells using the samples of lesional and uninvolved skin of four patients. Then, they collected the proteins secreted by the cells to the culture medium. They also labeled peptides with stable isotopes using a published protocol for dimethyl labeling [54]. The latter let them distinguish peptides produced in lesional and normal-looking skin. In turn, due to a high chromatographic resolution of the LC-MS/MS technique (Figure 2) [18], Williamson et al. were able to identify 59 secreted DEPs in lesional skin, including 36 previously unreported proteins (e.g., profilin 1/PFN1, galectin-related protein/GRP, and glutaredoxin-1/GRX1). As the level of PFN1 was significantly higher in the skin and blood serum of the patients compared to healthy volunteers, the authors proposed PFN1 as a biomarker of the disease [13].

Comparing the proteomes of the stratum corneum, Méhul et al. [26] identified ~140 DEPs of lesional and normal-looking skin and proposed some of them to monitor the flow of the disease. They also developed a new technique to assess the efficacy of topical treatments. Briefly, they collected proteins of the stratum corneum using adhesive strips. After extraction, the authors digested the recovered proteins. Then, they labeled the obtained peptides with isobaric tags, iTRAQs (Figure 3) [55], separated them by liquid chromatography and analyzed with qTOF mass-spectrometer. For reference, the term qTOF stands for “quadrupole time-of-flight” and refers to a certain way to accelerate ions during mass-spectrometry.

Based on the results of their study, Méhul et al. recommended noninvasive sampling of stratum corneum as a reliable method to quantify protein biomarkers in patients’ skin. As they also noticed, their protocol allowed ranking the drugs according to their clinical efficacy. Compared to normal-looking skin, skin lesions exhibited the elevated expression of chemoattractants of neutrophils (CXCL1 and CXCL8), T-cells (CCL4 and CXCL10), monocytes, and dendritic cells (CCL2, CCL4, and CCL20). Moreover, the authors proposed using CXCL1, CXCL8, CXCL10, and the soluble form of ICAM to monitor the progress of alternative topical treatments to rank them on their clinical efficacy [26].

Notably, the technique developed by Méhul et al. can also help in the differential diagnosis of psoriasis. In a short follow-up study [27], they analyzed the samples of stratum corneum obtained from patients with cutaneous T-cell lymphoma (CTCL) and psoriasis (*n* = 10 and 24, respectively). To the reference, cutaneous T-cell lymphoma is often confused with psoriasis because clinical manifestations of CTCL at the early stages are similar to ones in psoriasis. Analyzing the samples, the authors identified 431 and 543 in CTCL and psoriatic lesional skin, respectively, compared to normal-looking skin. As they noticed, both types of skin lesions shared many of these DEPs. However, IL37 and IL36G were significantly changed in psoriatic lesional skin and were not in patients with CTLC. Contrarily, CCL27 was differentially expressed in CTCL and was not in psoriatic skin lesions. Then, they established a molecular signature of 112 DEPs. This signature allowed them to distinguish the samples obtained from patients with CTCL and psoriasis. In addition, the authors validated the expression of 40 differentially expressed cytokines with the Luminex assay.

Swindell et al. [21] performed a comparative analysis of transcriptome and genome using paired skin samples of 14 psoriasis patients. Surprisingly, their study revealed a modest correlation between the expression levels of DEPs and their genes (r = 0.4) and concordant changes in limited numbers of DEPs and their genes (209 of 748 and 4122, respectively). In other words, significant changes in the expression of some genes might not cause significant changes at the protein level. In contrast, evident changes in the expression levels of some proteins might not correspond to significant changes in the transcriptome. For instance, this is correct for many DEPs involved in the translation of mRNAs, such as RPL3, RPS8, and RPL11. In addition, they discovered that psoriasis-associated DEPs and the others responded to different cytokines. The levels of the disease-associated concordant DEPs/DEGs (differentially expressed genes) could be changed by exposure of cultured keratinocytes to IL17A while the others could not. After all, their findings proved the primary role of IL17 in the regulation of disease-associated genes.

In a more recent paper, Szél et al. [36] demonstrated the ability of proteomics to discover new proteins previously unassociated with the disease. The comparative proteomic analysis of psoriatic and healthy skin performed by the authors revealed 249 DEPs. Comparing normal-looking skin to either lesional patients’ skin or skin of healthy volunteers, they identified ~30 proteins previously not associated with psoriasis. The authors also discussed DEPs that had unusual expression patterns. Primarily, they were interested in proteins that had bidirectional (contrasting) changes in their expression, namely ITGA7, ITGA8, PLVAP, PSAPL1, SMARCA5, and XP32. According to the authors, two identified DEPs, namely PRKDC and MYBBP1A, could be the potential key regulators of hyperproliferation and altered differentiation of skin cells, stress, and immune response associated with the disease.

Revealing the genetic vulnerabilities and exploring the main risks triggering the disease are among the principal problems addressed in population-specific studies. These studies appeal to a population of a particular country, people of certain races and ethnicity. In turn, their results help local authorities better define the people’s medical needs and timely provide the correct assistance. Because Asians are 5–6 times less susceptible to psoriasis than the Caucasian population of the industrialized western countries, the studies of people of Asian descent are one of the most valuable parts of population-specific studies. Being one of the largest Asian ethnic groups, the Chinese population represents the high interest in searching for the genetic variants that explain the resistance to the disease.

The authors of two complementing papers compared lesional skin of psoriasis patients to their normal-looking skin [49] and skin of healthy volunteers [40], respectively. In the first study, Wang et al. [49] analyzed six pools of samples (*n* = 15) labeled with TMT. The authors identified 3686 and quantified 3008 proteins. They found 102 and 124 DEPs with significantly higher and lower abundance in lesional skin, respectively (FCH > 1.5, FDR < 0.05). They also reported S100A9 and MMP3 as the most up- and downregulated proteins in their study (FCH—0.34 and 6.66, respectively). Comparing their data with previously published results, the authors suggested some populational differences between China and other countries. In this regard, they reported SART1 as DEP previously unassociated with the disease. In addition, Wang et al. [49] reported that they could not confirm the differential expression of GSTP1, SFN [21], KRT77, FLG2, and TREX2 [14] previously identified as DEPs by the others.

In the second study, Li et al. [40] reported multiple DEPs with altered expression in lesional skin of psoriasis patients (*n* = 11) compared to healthy volunteers (*n* = 11). The authors discovered an increased expression of proteins involved in the inflammatory response. Based on the results of their analysis, the authors suggested the activation of TNFα, IL17A, RAS, and NFκB signaling pathways. According to the authors, these changes explain the enhanced expression of proinflammatory factors, such as TNFα and various interleukins in skin lesions (e.g., IL8, IL18, and IL23). They also discovered the altered expression of the proteins induced by oxidative stress. Primarily, they mentioned myeloperoxidase (MPO) and monoamine oxidase B (MAOB). In addition, they reported nine DEPs previously not associated with the disease. Five of these proteins, namely MPO, TYMP, IMPDH2, GSTM4, and ALDH3A1, were upregulated. In contrast, four other proteins, namely CES1, MAOB, MGST1, and GSTT1, were downregulated. Discussing the possible role of new DEPs in the pathogenesis of psoriasis, the authors also emphasized their role in drug metabolism and ribosome biogenesis.

In a paper recently published by our lab [47], we focused on explaining the molecular basis of the disease. We discussed changes in the proteome that could be responsible for a higher metabolic rate in lesional skin. We also confirmed the previous discoveries of significantly more intensive translation in lesional skin [21]. We provided the data that explained the higher intensity of protein catabolism and membrane trafficking in lesional skin. Some changes in the proteome clarified how lesional skin accelerated the energy exchange and achieved a faster turnover of epidermal keratinocytes. We also suggested two scenarios that prepared the epidermis for remodeling: one—to modulate the developing inflammatory response in the extracellular space and another—to accelerate the protein metabolism in the cells affected by the disease.

Many authors, including us (e.g., [47,56]) emphasize that understanding key pathologic events in psoriasis requires the analysis of cells and pathways in normal-looking patients’ skin. Despite its vulnerability to the disease, normal-looking skin may have a protective mechanism preventing psoriasis from spreading up across the skin. Comparing normal-looking psoriatic skin to healthy skin, we showed that the kallikrein-kinin system was less active. We also discovered significant downregulation of ceruloplasmin/CP and apolipoprotein E/APOE in normal-looking skin. According to the others [57], these changes might cause suppression of VCAM-1 and ICAM-1. These changes were also needed to protect normal-looking skin from the invasion of activated immune cells.

A detailed study of disease-associated proteins revealed 123 DEPs in lesional and normal-looking skin of female patients. Contrarily, their expression did not significantly change in similar samples of male patients [48]. The following ontology analysis discovered 14 overrepresented biological processes. One of them was GO:0043627, response to estrogen. In contrast, a similar analysis of 172 DEPs of male patients that were not differentially expressed in similar samples of female patients did not reveal proteins involved in response to estrogen. The data validation with ELISA and qPCR performed on a larger cohort of female patients (*n* = 20) confirmed the differential expression of six proteins, namely HMOX1, KRT19, LDHA, HSPD1, MAPK1, and CA2. We proposed that these proteins could be a part of a protective mechanism that facilitates an adaptation to the progressing inflammatory response (HMOX1 and LDHA) as well as higher transcription (HSPD60) and proliferation (KRT19) rates. As we believe, the proposed protective mechanism (Figure 4) may include the cooperation of estrogen and hypoxia-induced factor 1α (HIF1α), which is highly unstable under normoxic conditions. However, HIF1α becomes more stable in lesional skin because lactate, the preferential product of LDH-M, stabilizes HIF1α by inhibiting prolyl hydroxylase (PHD2) [58]. It also triggers the nuclear translocation of HIF1α and the following induction of hypoxia-responsive genes [59], such as the six DEPs that we named above.

Because the results of similar proteomic studies that used various experimental approaches discovered significant data variability (compare [16] and [18]), Zhou et al. [41] wanted to define the most accurate way to obtain tissue samples. First, they used shaved biopsy instead of a punch biopsy. It was necessary to avoid sampling the subcutaneous fat. Second, they fragmented proteins using dispase to completely separate the epidermis and dermis. Third, they labeled the generated peptides with iTRAQ reagents. Comparing lesional skin to healthy skin (*n* = 16 and 15, respectively), the authors identified 269 DEPs among other 7269 proteins. Although Zhou et al. [41], reported many proteins already recognized by the others, they also discovered some DEPs, namely OAS2, IFIT3, IRF3, and MeCP2, that were not previously associated with psoriasis. Analyzing the expression of OAS2 in patients’ skin and blood, the authors discovered its strong correlations with PASI and BSA (r^2^ = 0.529–0.695). Thus, they proposed OSA2 has a role as a biomarker that can be used to monitor the disease and assess the therapeutic response.

More than half of the proteins encoded in the human genome undergo post-translational modification. Protein modifications influence their interactions with substrates and binding to other proteins, changes cellular location, signal transduction, enzymatic activity, and stability. To date, about two hundred protein modifications are known. Lysine 2-hydroxyisobutyrylation (Khib) is one of several known acetylations that target lysine residues (review in [60]). In histones, the sites of Knib are present at the N-termini. In addition, Knib targets lysine residues in their main globular domains. Respectively, Knib will likely disturb the interaction of histones with DNA and cause sterical obstacles in the nucleosomes. Knib also targets the enzymes of the pentose phosphate pathway, the cycle of tricarbonic acids, glycolysis, and gluconeogenesis [61]. In these enzymes, Knib will presumably interfere with substrate binding and catalysis. As Knib influences the activity of glycolytic enzymes, it would be reasonable to propose that Knib plays an essential role in the pathogenesis of psoriasis because psoriasis significantly accelerates the cellular metabolic rate in lesional skin. In contrast, Knib is one of the possible ways to slow it down.

In a recent study Ge et al. [35] analyzed the differences in the Khib profiles of 45 psoriasis patients. Comparing the patients’ lesional and uninvolved skin, the authors identified 94 differentially modified sites in 72 DEPs. Although the observed changes were bidirectional (51 sites were modified more frequently and 44—less frequently modified), the authors noticed significant decreases in the levels of the modified S100A9 (14%, *p* = 0.0004), S100A2 (37%, *p* = 0.02), and SERPINB4 (39%, *p* = 0.046) and the transporter of fatty acids FABP5 (44.5%, *p* = 0.067). Contrarily, the levels of tenascin (308%, *p* = 0.0307) and HSP90B1 (223%, *p* = 0.021) were significantly higher in lesional skin. Moreover, the ontology analysis revealed the highest representation of modified proteins in the P13K-Akt signaling pathway. Although the authors did not discuss it in the paper, their data indicate that the most modified proteins exhibited lower abundance in lesional skin than normal-looking skin of the patients and vice versa. Respectively, Knib might reduce the functionality of DEPs upregulated by the disease. At the same time, it likely produced the opposite effect downregulated DEPs.

In conclusion, the reviewed proteomic studies revealed a variety of biomarkers helpful in diagnosing and monitoring the disease. The main remaining challenges to proteomic studies are the complexity of the samples and, especially for skin, the high abundance of insoluble proteins. The role of many potential biomarkers in the pathogenesis of psoriasis remains unclear, and their mechanisms of action are not yet fully explored. As we show below, the constant improvement of techniques and instrumentation already help monitor the therapeutic response and predict the resistance to therapy. In the future, the broad implementation of disease-specific biomarkers to clinical practice will revolutionize our understanding of psoriasis and lead to the development of a personalized approach in medicine.

## 3. The Studies of Patients’ Blood

To date, histopathological analysis of skin specimens remains one of the most common and efficient clinical identification methods. However, this method is inconvenient for the patients as taking skin biopsies is invasive. Moreover, it is unlikely that skin specimens would be used to test people for predisposition to psoriasis before they present any skin manifestations. Respectively, there is a need for non-invasive molecular techniques that could help to accomplish these tasks [3].

For the last twenty years, proteomics is often used to identify DEPs in patients’ blood samples compared to blood samples of healthy volunteers and suggest biomarkers for clinical evaluation. Human blood contains essential information about many physiological and pathological conditions. As a reservoir of proteins secreted from different organs and tissues, the blood adequately reflects the status of human health. As psoriasis generally manifests as chronic skin inflammation, the concentration of proinflammatory cytokines in the patients’ plasma is significantly higher than in the plasma of healthy individuals. In this respect, studying the specific metabolic changes produced by psoriasis can be conducted using the patients’ blood. First, the collection of blood by venipuncture is minimally invasive. Second, combining venipuncture with one of the proteomic approaches provides access to relevant biomarkers of the disease. Third, comparing the blood samples taken at different time points allows precise monitoring of the therapeutic response.

Plavina et al. performed two studies to analyze peptides and proteins from the patients’ blood [12,13]. In the first study [12], the authors depleted the plasma of serum albumin and immunoglobulins. Then, they analyzed the plasma for the isoforms of glycoproteins using multi-lectin affinity chromatography (M-LAC) coupled with nano LC-MS/MS. Analyzing the patients’ blood, they identified numerous lower abundance proteins previously not associated with the disease. They also found that the concentration of some cytoskeletal and Ca^2+^-binding proteins and their peptides increased in the patients’ blood. The authors did not suggest any disease-specific biomarkers for psoriasis because the levels of identified proteins could vary more than 30-fold from patient to patient. However, they recommended their protocol for studies of potential glycoprotein biomarkers. As they noticed in the paper, their original protocol offered a good balance between throughput and sensitivity, and it is crucial to discover new biomarkers.

In the second study [13], Plavina et al. compared two protocols. The first protocol was M-LAC followed by nano LC-MS/MS. They had already used it in their previous study [12]. The second protocol included ultracentrifugation and bynano LC-MS/MS with electrospray ionization Fourier transform ion cyclotron resonance mass spectrometer (ESI-FTICR) [62]. Similar to qTOF, ESI-FTICR is a high-resolution and high-throughput method. Unlike qTOF and other types of spectrometers that measures *m*/*z* ratios of the produced ions, ESI-FTICR determines their frequencies and then, it transforms the frequency spectra to the *m*/*z* spectra.

The authors identified 21 DEPs, including thymosin β4, talin 1, γ-actin, filamin, profilin, S100A8/calgranulin A, and S100A9/calgranulin B. Most of the identified proteins were previously associated with different autoimmune and inflammatory disorders. In addition, four of them, namely calgranulins A and B, galectin 3, and galectin 3–binding protein, were either linked to psoriasis or altered in psoriatic serum. Although the results obtained by both methods were consistent with each other, the authors suggested using both protocols in parallel to reveal additional details on the potential blood-circulating biomarkers.

Fattahi et al. [19] identified four DEPs using two-dimensional gel electrophoresis coupled to MALDI/TOF-TOF. For reference, the term MALDI stands for matrix-assisted laser desorption/ionization. In other words, this type of mass spectrometer uses a laser to evaporate ions from the analyzed sample. Using a laser to ionize peptides prevents their excessive fragmentation because it does not break covalent bonds in the analyzed molecules. Using MALDI/TOF-TOF technology, the authors discovered significantly less retinol-binding protein (RBP4) in the patients’ blood than in the blood of healthy volunteers. They were the first who noticed a higher abundance of KRT10 in the patients’ blood, despite lower KRT10 expression in their lesional skin [47]. In addition, they reported two new isoforms of α1-antitrypsin/SERPINA1. These two proteins were present in the patients’ blood and were undetectable in the blood of healthy individuals.

To find the biomarkers that exhibit the strongest correlations with disease severity, Reindl et al. [22] compared the protein expression in the blood plasma of healthy volunteers and psoriasis patients. Using the LTQ-Orbitrap-nanoLC-MS/MS technique, the authors identified 208 DEPs and selected a panel of 57 potential biomarkers. They also validated 15 DEPs and suggested four of them, namely Zn-α2-glycoprotein/AZGP1, complement C_3_, polymeric immunoglobulin receptor/PIGR, and plasma kallikrein/KLKB1, to distinguish patients and healthy individuals. They also discovered eleven combinations of DEPs that might serve even better for the same purpose. In addition, the authors reported that the expression levels of desmoplakin/DSP, complement C_3_, PIGR, and KRT17 strongly correlated with patients’ PASI.

Following similar objectives, Matsuura et al. [25] analyzed the peptides of blood serum that could be associated with psoriasis. Using MALDI-TOF MS and Triple-TOF MS/MS, the authors compared the samples of blood serum obtained from the patients with psoriasis, psoriatic arthritis, and atopic dermatitis. The authors identified 93 differentially expressed peptides (DEPts) in the named groups of patients. The authors also found that many psoriasis-specific DEPts were fragments of four proteins, namely fibrinogen α/FGA, filaggrin/FLG, thymosin beta-4/TMSB4X, and FLJ55606 (highly similar to alpha-2-HS-glycoprotein). The authors proposed that the discovered DEPts, named p1466 (N-terminal alanine-defective fibroprotein A, FPA_2–16_) and p1977 (FLG_2099–2118_, Q_2099_pE, and Q_2115_E), could contribute to the development of the inflammatory response in the patients. As they observed in the following experiments with cultured cells, these peptides moderately influenced the secretion of humoral factors (GROα, IL-8, MCP-1, MIF, and SERPINE1) and cytokines (VEGF, pentraxin-3/PTX3, MIF, lipocalin-2/LCN2, osteopontin/OPN, and DKK1).

Using the SomaScan™ technology (SomaLogic, Boulder, CO, USA) [63] based on the specific interaction of the desired proteins and predesigned chemically modified oligonucleotides known as aptamers (Figure 5), Wang et al. [28] analyzed the profiles of serum proteins in four groups of individuals—patients with active psoriasis, atopic or contact dermatitis, and healthy individuals. The authors discovered four DEPs, namely kynureninase/KYNU, lectin galactoside-binding soluble 3-binding protein/LG3BP, and tryptase β2/TPSB2, and carbonic anhydrase 6/CA6, in the sera of psoriasis patients. They also found that the KYNU expression was significantly higher in psoriasis patients than the patients with atopic or contact dermatitis and healthy individuals. Wang et al. also noticed that the KYNU level in the serum let them distinguish the individuals with exacerbated psoriasis from other patients. Previously, several transcriptomic studies [64,65,66] identified *KYNU* as a DEG in psoriatic skin. Moreover, KYNU is also synergistically regulated by proinflammatory cytokines IL17 and TNFα that play a central role in the pathogenesis of psoriasis [67]. Although Wang et al. did not find any significant correlation between KYNU level and PASI, the authors suggested KYNU as a biomarker for psoriasis.

Recently, Gęgotek et al. [29] compared blood plasma donated by psoriasis patients and healthy volunteers. In their study, the authors used nanoflow LC-MS/MS combined with a Q-Exactive OrbiTrap mass spectrometer to analyze in-gel digested peptides. The authors identified 486 proteins. They also discovered that patients’ plasma contained significantly less vitamin D and several proteins involved in lipid metabolism (e.g., apolipoprotein M/APOM). Contrarily, the plasma levels of proteins involved in immune response (IL6, IL23, anti-factor VIII, and immunoglobulin GCT-A3) and signal transduction (POTE ankyrin domain family member F/POTEF, AT motif binding factor1/ATBF1, and trimethylguanosine synthase/TGS1) were higher in plasma of psoriasis patients. They also reported an increased expression of proteins directly involved in the secretion of signaling molecules (biotinidase/BTD and BAI1-associated protein 3/BAIAP3). In addition, the authors discussed the influence of lipid peroxidation on the proteomic profile of psoriasis patients. They showed that a higher intensity of lipid peroxidation in the patients’ blood accelerated the formation of protein adducts and increased their complexity compared to the blood of healthy volunteers.

Immune cells, primarily the lymphocytes, which infiltrate psoriatic skin, are one of the principal sources of proinflammatory cytokines. These cells secrete koebnerisin/S100A15 and psoriasin/S100A7) [68]. Then, koebnerisin and psoriasin synergistically induce proinflammatory cytokines (IL1b, TNF, IL6, and IL8) in the immune cells. Their induction, in turn, accelerates the inflammatory response promoting the disease progression and development of comorbidities, such as cardiovascular disorders [69]. In contrast, the suppression of koebnerisin in the circulating lymphocytes reduces the expression of proinflammatory cytokines. The latter suggests that koebnerisin can serve as a biomarker to monitor the therapeutic response in psoriasis patients [68].

Exploring the proteomic profiles of epidermal keratinocytes, lymphocytes, and dermal fibroblasts of psoriasis patients and healthy volunteers, Łuczaj et al. found differences in the expression patterns of thioredoxin and glycolytic enzymes [70]. In patients’ keratinocytes and lymphocytes [34], they discovered a lower expression of TXNRD1, a protein with antioxidant activity. They also showed a higher expression of the glycolytic isoenzymes, namely PGAM1 and -2, converting 1,3-bisphosphoglycerate to 2,3-bisphosphoglyceratein. On the other hand, the authors discovered a higher abundance of TXNRD1 and a lower abundance of several glycolytic enzymes (PK, PGK2, ALDOL, and GAPDH) in dermal fibroblasts [39]. We presume that these results reflect the different physiological conditions in where the mentioned cells reside and their metabolic rates. In lymphocytes and keratinocytes, the metabolism accelerates because of their exposure to proinflammatory cytokines [29,34]. In turn, the differential expression of TXNRD1 indicates different rates of lipid peroxidation, which increases with the progression of the inflammatory response. The latter agrees with the authors’ discovery of higher levels and variety of 4-hydroxynonenal protein adducts in patients’ keratinocytes, lymphocytes, and blood sera [29,34].

In turn, the study conducted by Li et al. [31] provides a comprehensive proteomic signature of psoriasis in peripheral blood mononuclear cells (PBMC). The authors performed LC-MS/MS analysis of TMT-labeled samples that led to the identification of 442 proteins with an altered expression more than 1.2-fold, compared to the healthy control. To the reference, the abbreviature TMT stands for “tandem mass tags”. The TMT-labeling protocol exploits the same principle as iTRAQ that we discussed above. Briefly, the mixes of digested proteins (peptides) conjugate with small amino-reactive isobaric molecules that become separated from their targets during mass spectrometry. As 16 different TMT labels are currently available on the market, up to 16 samples can be pulled and analyzed at once. Unlike the others, Li et al. [31] worked with the cells obtained from patients with new-onset psoriasis (*n* = 31) who had noticed the first signs of the disease no longer than 30 days before the blood collection. The latter explains their ability to identify many new disease-associated proteins (ATM, SLFN5, ZNF512, SPATA13, DOCK2, ARSB, VIRMA, NRGN, etc.).

Summarizing the previous findings on psoriasis biomarkers in the human sera, Xu et al. [37] developed a protein array of 129 specific antibodies to the protein biomarkers of the disease previously suggested by the others. In brief, the authors printed their specific antibodies on a 3D-modified slide. They also included appropriate negative and positive controls to the array. Then, they validated the biomarkers with sera samples donated by healthy volunteers and psoriasis patients. Comparing the expression levels of sera proteins in both groups, the authors showed the relevance of 112 proteins to the disease. They also found statistical differences in the expression of nine proteins that belonged to the array, namely PI3, TNFRSF8, PFN1, KRT16, TNFSF8, KLK1, APOC3, CXCR3, and CCL4. Following the DIA-MS approach that presumes a second MS analysis following additional fragmentation of the identified peptides, Xu et al. discovered 58 DEPs. They also validated them with their custom-made array of antibodies. The obtained results indicated that the expression of SERPINE1, PI3, IL4, and CX3CL1 correlated with the number of neutrophils in the blood. In turn, the expression of PI3 (r = 0.85, *p* < 0.01), CCL22, and IL12B discovered a strong correlation with PASI. Moreover, the expression of TNFRSF8 and CD14 correlated with the VAS score, i.e., a quantitative assessment of itch intensity. In addition, Xu et al. [37] split their patients into two groups. Patients of the first group responded to the traditional herbal medicine YinXieLing used in China to treat psoriasis; patients of the second group did not respond to the therapy. Comparing the groups, the authors discovered three predictive biomarkers of the favorable outcome for the treatment, namely FCN2, MIF, and MMP1.

In summary, various proteomic studies have identified more than 100 potential biomarkers in patients’ blood. On the other hand, the diversity of psoriasis complicates their validation, and the similarity of different autoimmune disorders makes it difficult to prove their relevance to a particular disease. Future studies of potential biomarkers are needed to explain correlations of their levels and severity of psoriasis by connecting their mechanisms of action and biological activities to the pathogenesis of the disease.

## 4. Clinical Applications

### 4.1. Monitoring the Therapeutic Response

Psoriasis is known for its complexity and heterogeneity. The complexity of psoriasis presumes that the disease alters the expression of multiple genes and causes serious miss-regulation of various signaling pathways. Moreover, the disease causes pathological changes in many tissues and organs. Heterogeneity of psoriasis appears in multiple skin phenotypes and associated comorbidities that significantly affect the patients’ quality of life. Although many treatment options are available, the cure for psoriasis is yet not found. Up to a third of psoriasis patients do not respond well to the prescribed therapies. On the other hand, patients non-responding to one of the biologics can experience an improvement after switching to another [71].

The first attempt to use a proteomic method to monitor the disease response to a therapeutic agent was performed by Bonnekoh et al. in 2007 [11]. In their study, the authors wanted to identify the specific biomarkers of therapeutic response to efalizumab, a recombinant humanized monoclonal antibody that neutralized the CD11a subunit of lymphocyte function-associated antigen 1 (LFA-1). Using multi-epitope ligand cartography (MELC), based on an array of specific antibodies, they compared the protein expression in lesional and normal-looking skin of psoriasis patients before and after the treatment. Analyzing the data obtained before the treatment and after the completion of therapy (12 weeks later), Bonnekoh et al. in 2007 [11] identified several groups of potential biomarkers. For instance, they found that the level of pan-cytokeratin correlated with epidermal thickness among the responders to efalizumab (*n* = 5). They also showed that syndecan 1/CD138 and transferrin receptor protein 1/TRF1/CD71 expression could serve as prognostic factors in the assessment of treatment efficiency. In addition, they discovered that the levels of CD45, CD2, CD4, and CD8 was significantly decreased in responders suggesting that these DEPs could monitor the presence of leukocytes and different subpopulations of T-cells.

Kolbinger et al. [24] compared the expression of 170 proteins in the blood and derma of psoriasis patients and healthy individuals before and after the treatment with the fully human anti-IL17A monoclonal antibody, secukinumab. Using the method of proximity extension assay developed by Olink, Sweden (Figure 6), the authors found that the expression of antimicrobial peptides, proinflammatory cytokines, and neutrophil chemoattractants significantly increased. Then, their expression returned to normal levels after the treatment with secukinumab. Based on the obtained results, the authors recommended using β-defensin 2/DEFB4 to monitor IL17A-driven pathology and the therapeutic response to secukinumab in serum and dermis.

Foulkes et al. used SomaScan [32] to explore the serum proteome of psoriasis patients (*n* = 10) before and after the treatment with etanercept. The authors discovered a higher expression of TNF-dependent proteins in patients’ blood. They also saw an altered expression of proteins belonged to the interferon signature that normalized after the treatment. For reference, a similar response to etanercept was previously discovered in the skin [72].

The study performed by Medvedeva et al. [38] clarified the connection between the severity of psoriasis, drug-induced pharmacodynamic effects, and the patient’s response status. The authors analyzed 150 plasma samples obtained from psoriasis patients in a phase III clinical study for apremilast, a specific inhibitor of phosphodiesterase 4 (PDE4) using the method of proximity extension assay. They identified IL17A and KLK7 as biomarkers for disease severity and apremilast pharmacodynamic effects in psoriasis patients. They also found that the combined expression rate of four DEPs, namely KLK7, PEDF, MDC, and ANGPTL4, significantly declined in responders compared to non-responders to apremilast. As a result, they recommended to use the combined expression rate of KLK7, PEDF, MDC, and ANGPTL4 in the patients’ blood to identify responders to apremilast among psoriasis patients.

In summary, the development of therapeutic antibodies and small molecules targeting psoriasis into clinical practice resulted in significant improvement for patients with severe psoriasis. Although the previously performed proteomic studies have allowed to identify some biomarkers of the therapeutic response in inflammatory skin diseases [11,73], the development of clinical applications that would use their results is still far from completion [74]. Some discovered biomarkers are not specific enough because they are similarly altered in several diseases. The role of the others in the pathogenesis of psoriasis remains unclear. In this regard, there is a persistent demand for additional studies that could resolve these two important problems.

### 4.2. Drug Evaluation

Although psoriasis is considered a skin disease, the patients often experience comorbidities, such as cardiovascular diseases, metabolic syndrome, inflammatory bowel disease, etc. [75]. Moreover, the systemic treatment of psoriasis commonly causes side effects. For these reasons, the timely diagnostics of unwanted metabolic changes and organ dysfunction as early as possible is highly desired. The existing proteomic approaches can potentially diagnose the mentioned health conditions and scale the actual damage to the organs and tissues. Respectively, their results will clarify the details of the personal treatment plan for the patient.

Patients with psoriasis more frequently experience myocardial infarction, stroke, and coronary heart disease [76]. To assess the contribution of antipsoriatic medications to cardiovascular events, Kim et al. [30] analyzed the data of phase III clinical trial NCT01241591 [77] for etanercept and tofacitinib. For reference, etanercept is a fusion protein of TNF receptor 1 (TNFR1) and F_c_ domain of IgG_1_. It binds and neutralizes TNF, one of the key proinflammatory cytokines in psoriasis [78]. Tofacitinib is a small molecule that inhibits Janus kinases JAK1 and JAK3 that mediate the inflammatory response via activation of STAT proteins [79]. Both drugs target the inflammatory immune cells, primarily T-cells, via different pathways. The data analyzed by Kim et al. were obtained by the method of proximity extension assay [80]. In the samples obtained before the treatment, the authors discovered the highest correlation of PASI with the expression of four genes, namely IL17A and –C, IL20, and CCL20, among the patients responding to both medicines [30]. In the samples obtained after the completion of three-month therapy, they showed a positive correlation of PASI and the expression levels of IL17A and –C among the responders to tofacitinib (r = 0.24 and 0.23, respectively). Contrarily, the responders to etanercept did not discover a significant correlation. Moreover, they found that tofacitinib suppressed more cardiovascular blood proteins than etanercept. Thus, Kim et al. concluded that tofacitinib was a more efficient suppressor of blood proteins associated with cardiovascular events. They also noticed that tofacitinib was more beneficial for psoriasis patients with a high risk of cardiovascular events responding to the therapy.

### 4.3. Discovering Risk Factors of Comorbidities and Their Analysis

As we have already mentioned above, psoriasis patients are often diagnosed with comorbidities. At some point in their life, ~30% of patients develop psoriatic arthritis. In contrast, ~15% of patients with arthritis develop psoriasis [56]. In addition, the term “psoriatic arthritis” is inclusive of peripheral spondyloarthritis, enthesitis, and dactylitis [81]. Presently, there is a need for a clinical test to identify individuals who already suffer from one disease and can be predisposed to another.

To achieve this goal, Cretu et al. [20] performed a pilot LC-MS/MS study of paired skin samples donated by psoriasis patients with and without psoriatic arthritis (2 groups of 10 patients). The authors identified 47 proteins differentially expressed in either lesional or normal-looking skin. Then, they selected eight DEPs, namely SRP14, ITGB5, POSTN, SRPX, FHL1, PPP2R4, CPN2, and GPS1, with elevated expression in either phenotype of patients and validated them by ELISA on sera samples donated by another group of patients (15 individuals with psoriatic arthritis and 33—without it). Based on the results of the validation analysis, the authors proposed integrin β5/ITGB5 as a candidate biomarker of psoriatic arthritis.

Because Cretu’s study had two significant limitations: the researchers pooled samples (*n* = 5) before analyzing them by LC-MS/MS, and only two proteins, namely ITGB5 and POSTN, were validated by ELISA, other attempts to identify the biomarkers of psoriatic arthritis followed. Recently, Leijten et al. [45] used the proximity extension assay (see above) to identify sera proteins with expression levels strongly correlating to major clinical features of psoriasis and psoriatic arthritis. The authors analyzed the samples of blood sera donated by psoriasis patients (*n* = 18), patients with psoriatic arthritis (*n* = 20), and healthy volunteers (*n* = 19). Comparing the protein expression in psoriasis patients and patients with psoriatic arthritis, the authors failed to identify proteins with altered expression in one of the groups. At the same time, they found that ICAM-1 and CCL18 had the strongest positive correlation with the number of swollen joints (r = 0.81 and 0.76, *p* < 0.001, respectively) while PI3 and IL17RA exhibited the strongest correlation with PASI score (r = 0.54, and −0.51 *p* < 0.01, respectively). They also reported that both groups of patients were separable by hierarchical clustering. The latter suggested the existence of a hidden molecular signature characteristic for psoriasis patients predisposed to psoriatic arthritis, despite that the authors did not report it in their paper. Thus, theoretically, it may be possible to identify people with a predisposition to psoriatic arthritis among psoriasis patients. Respectively, we may be able to timely and efficiently advise these patients and control their disease.

Unlike their predecessors focused on circulating biomarkers in the bloodstream, Zue et al. [50] compared PBMCs of psoriasis patients with and without psoriatic arthritis to PBMCs of healthy volunteers (4 samples in each group). The authors explained their choice for two reasons. First, PBMCs are easy to get. Second, they are more stable than plasma proteins. In addition, we would also acknowledge that at the early stages of psoriatic arthritis, when symptoms of the disease are not yet evident, the patients should already have specific immune cells homed to their joints. In turn, these cells would likely contain the disease-associated biomarkers at higher levels than the same kind of biomarkers in the patients’ blood.

After iTRAQ-labeling, the authors analyzed the samples by LC-MS/MS and discovered 60 proteins with differential expression between two groups of patients (*p* < 0.05). According to the authors, 14 proteins, namely SIRT2, NAA50, ARF6, ADPRHL2, SF3B6, SH3KBP1, UBA3, SCP2, RPS5, NUDT5, NCBP1, SYNE1, NDUFB7, and HTATSF1, can be potential biomarkers of psoriatic arthritis. These proteins were differentially expressed in psoriasis patients with psoriatic arthritis and were not in patients without it. Although the authors minimized the number of samples used in their MS study, they validated their results on the best candidate biomarker SIRT2 by Western blot on a larger group of participants (*n* = 21). Based on the results of their analysis, Zue et al. [50] concluded that SIRT2 could serve as a biomarker of psoriatic arthritis in psoriasis patients.

Large-scale population studies suggest that psoriasis increases the risk of cardiovascular disease (CVD) by upwards of 50% [82]. Psoriasis is also an independent risk factor for cardiovascular mortality [83]. The elevation of proinflammatory cytokines, such as IL17, IL23, TNF, and IFN-γ, enhances vascular inflammation that, in turn, causes dysfunction of the cardiovascular system. The proinflammatory cytokines produced by the inflamed vascular endothelium attract the immune cells promoting endothelial dysfunction, impaired insulin sensitivity, and increased carotid intima-media thickness and aortic stiffness [69].

Recently, Elnabawi et al. [42] proposed a computer model that can assess the risk of cardiovascular disease in psoriasis patients. Using proximity extension assay, they compared the changes in the proteomes of brachial vein endothelial cells of psoriasis patients (*n* = 23) and healthy volunteers (*n* = 10). The authors analyzed the expression of 273 proteins associated with the inflammatory response. They identified 17 differentially expressed proinflammatory cytokines, including IL17A, IL6, and CCL20. Their levels were significantly higher in the patients’ blood. Analyzing the data, Elnabawi et al. [42] discovered a correlation between LDL-cholesterol and the expression of circulating CCL20 and IL6. They also found that the expression level of CCL20, which is a chemoattractant of dendritic and immune cells [84], strongly correlated with vascular endothelial inflammation score (r = 0.55, *p* < 0.001) and PASI (r = 0.57, *p* = 0.01). The addition of CCL6 as an independent variable to the original computer model increased its specificity and efficiency. Respectively, the authors recommended CCL6 as a biomarker of impaired vascular health in psoriasis patients.

Moreover, the same group also reported [33] of a proinflammatory molecular signature composed of eight cytokines, namely IL1β, CXCL10, VCAM-1, IL-8, CXCL1, LTB, ICAM-1, and CCL3, to describe the changes in the patients’ brachial veins endothelial cells accompanying atherosclerosis. The expression of the identified cytokines exhibited similar changes in the serum, brachial veins endothelial cells, and lesional skin of psoriasis patients. Moreover, their expression correlated with disease severity. The authors concluded that they detected a coordinated response among the proinflammatory cytokines. They also proposed that an identified mechanism regulates pathological changes caused by psoriasis in different tissues, such as the skin and blood vessels.

Using the same experimental approach, Kaiser et al. [44] compared the plasma proteomes of 85 patients diagnosed with moderate to severe psoriasis with or without established atherosclerotic cardiovascular disease. Among the discovered DEPs, they found biomarkers of atherosclerosis previously identified by the other methods. They also showed that GDF15 expression negatively correlates with vascular inflammation in the ascending aorta (r = −0.47, *p* < 0.01) and entire aorta (r = −0.44, *p* < 0.01). Moreover, it positively correlated with carotid intima-media thickness (r = 0.53, *p* < 0.01) and coronary artery calcium score (r = 0.40, *p* = 0.018) in psoriasis patients without cardiovascular disease and statin treatment. In this regard, the authors concluded that the GDF15 level in the patient’s blood could serve as a biomarker of subclinical atherosclerosis in patients with psoriasis.

As the changes in the proteome of psoriasis patients are well-documented, psoriasis may serve as a model in comparative studies that explore the molecular bases of other chronic inflammatory disorders. To date, several papers have compared the inflammatory potentials of psoriasis, atopic dermatitis [23], alopecia areata, [43], hidradenitis suppurativa [46,51] using the proximity extension assay. The authors of the mentioned studies wanted to exploit the opportunity of using the same treatment options for the diseases with similarities in the inflammatory response and assess the risk of cardiovascular events in different groups of patients. Notably, these studies pursued the same goal, used the same experimental approach and reagents, and applied the same criteria to select DEPs (FCH > 1.2–1.3, FDR < 0.1).

Brunner’s paper [23] was the first study that defined and characterized inflammatory and cardiovascular risk proteins commonly upregulated in atopic dermatitis and psoriasis. Analyzing the blood samples donated by patients with moderate-to-severe atopic dermatitis (*n* = 59), psoriasis (*n* = 22), and healthy volunteers (*n* = 18), the authors discovered significant differences between the two groups of patients. They showed upregulation of many identified cardiovascular biomarkers in either group of patients (in both cases, they identified 48 DEPs). This finding suggested that both disorders increase the risk of cardiovascular events. Ten mutually miss-regulated proteins included the markers of Th_1_ (IFN-γ, CXCL9, and LTA) and Th_17_ (CCL20 and IL17C) immune responses, and several other proteins, namely IL2RA, IL16, IL20, BLMH, and CDCP1. In patients with atopic dermatitis, the authors found a significantly higher abundance of DEPs involved in Th_1_ (CXCL10, CXCL11), Th_2_ (IL-13, CCL13, CCL17, CCL11, IL-10), Th_17_/Th_22_ (S100A12), and Th_1_/Th_17_/Th_22_ (IL12/IL23p40) signaling pathways. This finding suggests a higher inflammatory potential of atopic dermatitis over psoriasis. In addition, the authors showed a significantly higher expression of proteins associated with atherosclerosis (CX3CL1, CCL8, M-CSF, and HGF) in the same group of patients. Respectively, the authors concluded that although both diseases share cardiovascular risk factors, such as arterial hypertension, diabetes mellitus, and hypercholesterolemia, patients with atopic dermatitis exhibit a stronger systemic inflammation compared to psoriasis patients.

The study conducted by Glickman et al. [43] revealed a systemic nature of alopecia areata and its advanced forms alopecia totalis and alopecia universalis. The authors characterized the expression of proinflammatory and cardiovascular biomarkers in the blood of patients with moderate-to-severe alopecia areata (*n* = 35), atopic dermatitis (*n* =49), moderate-to-severe psoriasis (*n* = 19), and healthy volunteers (*n* = 36). Comparing three groups of patients, the authors revealed similar miss-regulations of immune, cardiovascular, and atherosclerosis biomarkers. They found that patients with the advanced forms of alopecia areata exhibited the highest systemic inflammatory tone, and higher expression of cardiovascular risk biomarkers (e.g., OLR1, OSM, MPO, and PRTN3), compared to the other groups of patients as well as their fellow groupmates without total involvement (30 < SALT < 95).

The point made by the authors that alopecia areata is a systemic disorder may potentially revolutionize the treatment of the disease. The data presented in this paper justify the necessity of the systemic approach for the therapy of alopecia areata and its advanced forms. The proposed benefits of systemic treatment may include but are not limited to lowering the frequency of cardiovascular events. For instance, it will improve the control of comorbidities and the patients’ quality of life.

In the two subsequent studies, the authors compared the blood [46] and the skin [51] of psoriasis patients, patients diagnosed with hidradenitis suppurativa, and healthy volunteers. Analyzing the sera samples, Navrazhina et al. [46] showed that hidradenitis suppurativa exhibited a significantly more intense inflammatory burden and an increase in cardiovascular/atherosclerosis-related biomarkers than psoriasis. The authors also found that disease severity (PASI—for psoriasis and ISH4—for hidradenitis suppurativa) was robustly correlated with the expression of hundreds of proteins involved in immune response and biomarkers of atherosclerosis (r^2^ > 0.25, *p* < 0.1). They also proposed a simple linear model that distinguishes both groups of patients using only two variables, namely the expression levels of peptidase inhibitor 3 (PI3) and lipocalin 2 (LCN2).

Examining the skin samples, Navrazhina et al. [51] found that chronic skin inflammation in the patients with hidradenitis suppurativa extended far beyond the skin lesions and sustains on a comparable level. The authors discovered significant upregulation of proinflammatory cytokines, namely TNF, CCL20, IL-6, IL-8, and IL-12B (FCH > 1.2; FDR < 0.05). In contrast, they did not observe similar changes in normal-looking skin of psoriasis patients. In lesional skin, the authors found a significant increase in the expression of proteins involved in Th_1_ (IL8, CCL3, CCL4, CXCL9, CXCL10, and CXCL11), IL12/IL23 (CCL3, CXCL9, TNF, IL-17A, and IL12B), and Th_17_ (CXCL1, CCL20, and IL17A) inflammatory responses in both groups of patients. In addition, they discovered a significantly higher abundance of proteins associated with cardiovascular events (EN-RAGE, OSM, TNF, MMP1, and IL8) in their lesional skin. The authors showed a significantly higher expression of proinflammatory cytokines, namely, TNF, CCL20, IL6, IL8, and IL12B (FCH > 1.2; FDR < 0.05), compared to healthy control. In contrast, they did not observe similar changes in normal-looking skin of psoriasis patients.

The results of data analysis performed by the authors allow considering hidradenitis suppurativa as a systemic disorder. Navrazhina et al. noticed that cytokines associated with IL12/IL23 signaling pathway, Th_1_, and Th_17_ immune responses, as well as biomarkers of atherosclerosis, are upregulated in both patients’ blood and skin [46,51]. Moreover, they found evidence of sustained inflammation in patients’ normal-looking skin. Respectively, the authors recommended treating the patients with hidradenitis suppurativa beyond skin nodules with systemic medicines. Relying on the similarities in both groups of patients, they also proposed to check the efficiency of approved anti-psoriatic systemic drugs for hidradenitis suppurativa. The latter, in their opinion, might significantly extend the list of treatment options available to patients.

### 4.4. Assessment of Adverse Effects

Long-term consumption of methotrexate (MTX), which is one of the most frequently prescribed drugs for psoriasis, causes severe adverse reactions, such as liver injuries and the following hepatic fibrosis [85]. At the same time, the conventional biomarkers for liver injury, such as plasma alanine aminotransferase (ALT), may not be conclusive to predict hepatic fibrosis in psoriasis patients treated with MTX [86]. A histological evaluation of liver biopsy, which represents a golden standard for monitoring MTX-induced hepatic fibrosis, causes a risk for the patients [87].

To identify specific biomarkers of MTX-induced hepatic fibrosis visible at an early stage of fibrosis, van Swelm et al. [17] compared urinary proteomes of psoriasis patients with and without liver fibrosis (*n* = 60) using MALDI-TOF MS. They divided the patients into two groups depending on whether patients exhausted their high cumulative MTX doses (less and more than 1500 mg MTX, respectively). The authors discovered that patients’ urine contained the precursors of serum albumin and cathepsin B (p-ALB and p-CATHB, respectively). They also found haptoglobin (HP), prostaglandin-H2 d-isomerase (PTGDS), inter-α-trypsin inhibitor heavy chain H4 (ITIH4), precursor of transferrin (p-TF), zinc-α2-glycoprotein (AZGP1), apolipoprotein D (APOD), cadherin E (CDH1) and cadherin N (CDH2) in patients’ who exhausted they high cumulative MTX dose. The authors reported that p-ALB and p-CATHB were absent in healthy control. At the same time, the other named proteins were undetectable in the samples of patients who consumed less than 1500 mg MTX. As only the levels of ITIH4 and CDH2 were significantly different between the two groups of patients, Swelm et al. [17] suggested these two proteins as potential biomarkers of MTX-induced hepatic fibrosis.

## 5. Conclusions

To date, both research and clinical scientists know about the potential benefits of the proteomic approach (MS and alternative technologies). The established protein biomarkers of psoriasis are suitable for the diagnostics of the disease, monitoring the therapeutic response, and predicting the treatment outcome [53]. For the next step, the proteomic analysis will become a part of multidisciplinary studies integrating phenotyping and multiomic data that represent individual patients. Many of these studies are in the planning stage, and the others are already in progress (e.g., [88,89]). Their results will associate specific genetic profiles of the patients with possible therapeutic incomes, assess the differences among psoriasis patients, and link these differences to certain habits in the patients’ lifestyles. They will also help clarify the interactions of psoriasis and comorbidities (cardiovascular disease, psoriatic arthritis, etc.) at the molecular level.

## Figures and Tables

**Figure 1 biomedicines-10-00619-f001:**
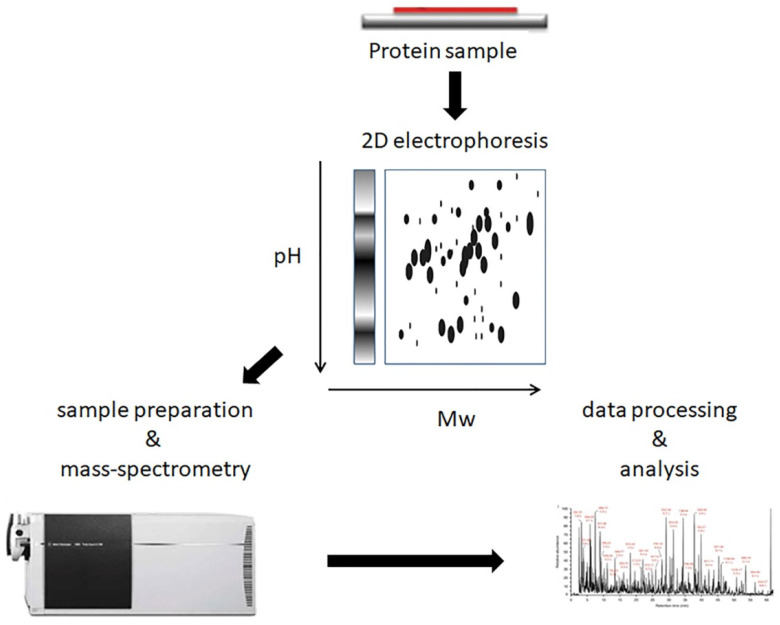
2D-gel electrophoresis followed by mass spectrometry. The experiment starts from electrophoretic separation of protein samples in two directions, by isoelectric point and by molecular weight (M_w_). Digestion in- or out-of-gel, mass spectrometry, and data analysis follow electrophoresis.

**Figure 2 biomedicines-10-00619-f002:**
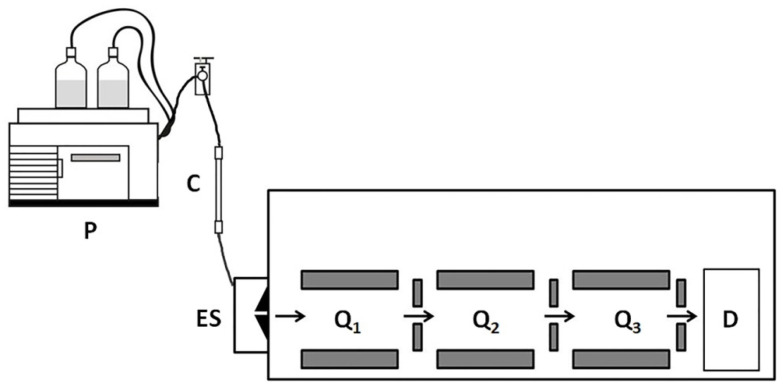
Liquid chromatography coupled to tandem mass spectrometry (LC-MS/MS). The experiment begins with a protein separation by liquid chromatography (LC) followed by ionization and mass spectrometry analysis. Triple quadrupole mass spectrometer comprises of three quadrupoles Q_1_, Q_2_, and Q_3_. Quadrupoles Q_1_ and Q_3_ serve as mass filters. The middle quadrupole Q_2_ functions as a collision cell. The specific precursor ions isolated by Q_1_ and accumulated in Q_2_ collide with neutral gas molecules, such as nitrogen generating the product ions. The product ions reach the third quadrupole assembly Q_3_. The quadrupole Q_1_ allows ions with a specific *m*/*z* ratio range to pass to the detector (D). P—pump; C –column; ES—electrospray, a liquid-to-aerosol converting device.

**Figure 3 biomedicines-10-00619-f003:**
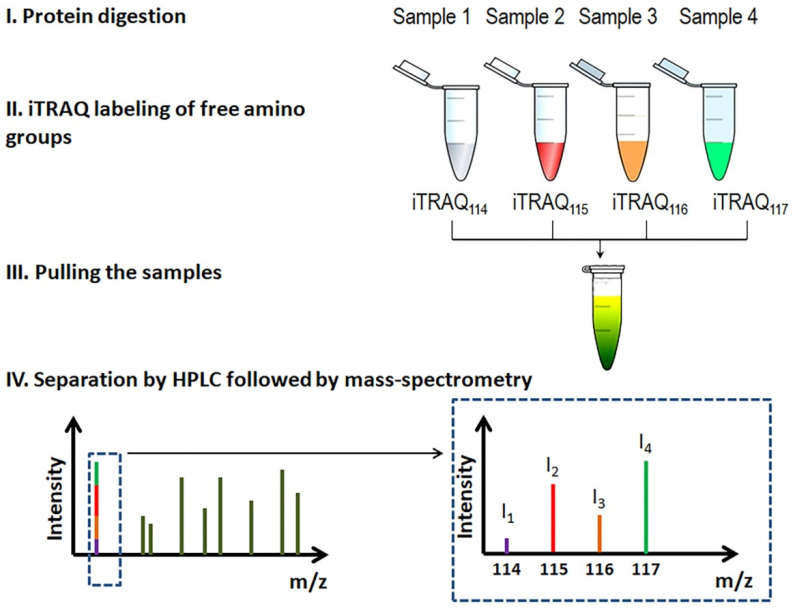
iTRAQ workflow. Several samples are separately digested and labeled with iTRAQ tags of different molecular weights and pulled. The labeled peptides are separated by HPLC followed by mass spectrometry. During mass spectrometry, tags become released from the peptide. The ratio of the iTRAQ ions is used to assess the sample-specific content of a particular peptide.

**Figure 4 biomedicines-10-00619-f004:**
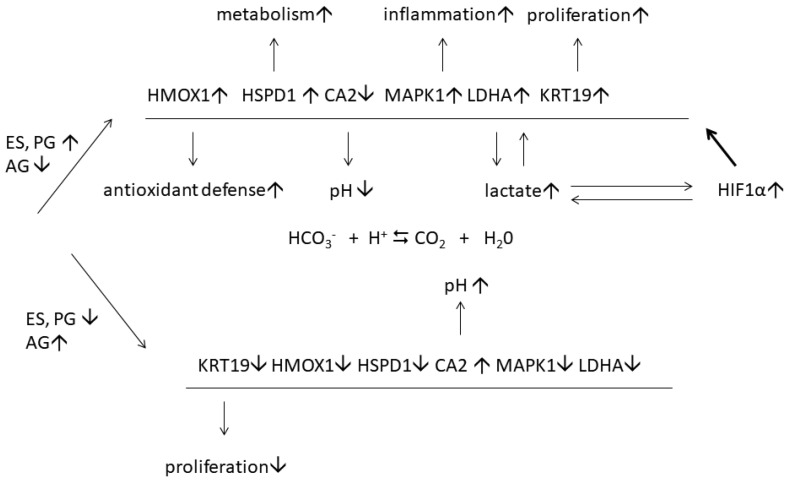
The regulation of estrogen-responsive proteins in lesional skin of female psoriasis patients. The signs ↓ and ↑ indicate that the protein expression is increased and decreased, respectively. The sign ⇆ indicates that chemical reaction between HCO_3_^−^ and can be reversed [48].

**Figure 5 biomedicines-10-00619-f005:**
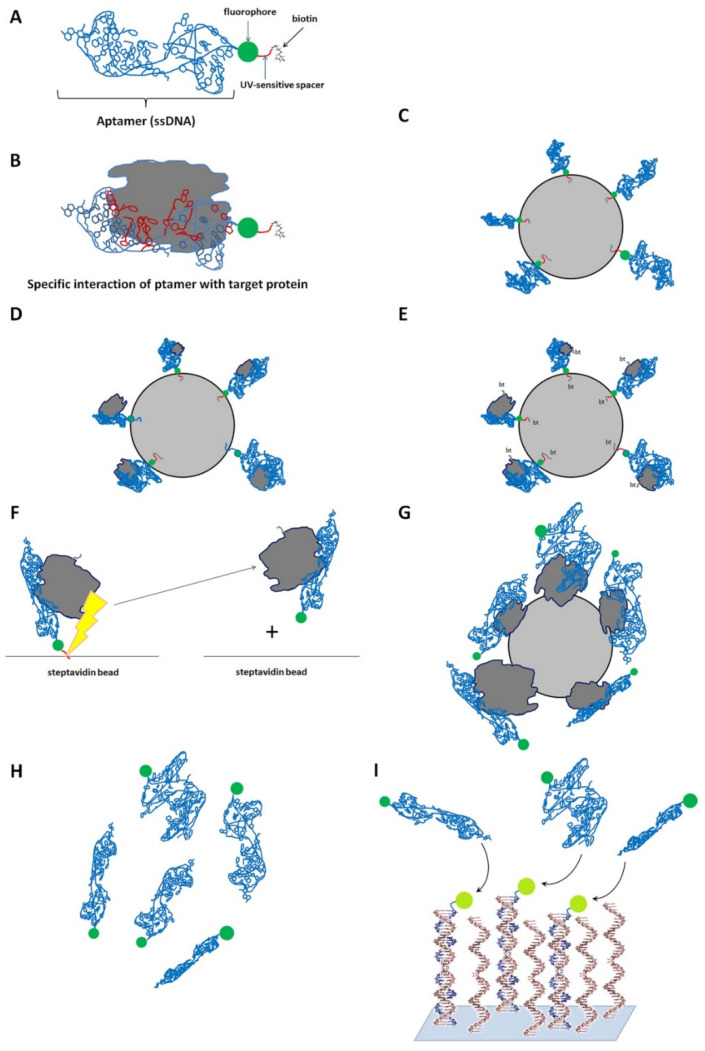
The principle of the SOMAscan. (**A**). Aptamers used in the study interact with a specific protein (target). Each aptamer contains three parts: a single-stranded DNA made of both canonical and modified deoxyribonucleotides (light blue), fluorophore (green), a UV-sensitive spacer (red), and biotin (grey). (**B**). The specific interaction of an aptamer with a target protein. The nucleotides directly interacting with a protein are shown in red and the others—in dark blue. A SOMAscan includes several steps. (**C**). Incubation of aptamers with streptavidin beads (Step 1). Because the aptamers contain biotin, they bind to the beads via biotin. (**D**). Incubation of streptavidin-coupled aptamers with the desired protein sample (Step 2). As protein samples used in the experiment presumably contain target proteins, aptamers specifically interact with their target protein in the 1:1 ratio. (**E**). Biotinylation of the target proteins with biotin, bt (Step 3). (**F**). Photosensitive cleavage of the spacers from aptamers followed by a separation of aptamers from the beads (Step 4). (**G**). The released complexes of aptamers and biotinylated proteins become recaptured by streptavidin beads (Step 5). As target proteins are biotinylated, they interact with streptavidin beads via biotin. (**H**). Dissociation of aptamers from their target proteins (Step 6). (**I**). Hybridization of aptamers with the probes of complementary single-stranded DNA attached to a slide (Step 7). Since each aptamer contains a fluorophore, the hybrid dsDNA molecules emit a fluorescence. The intensity of emitted light is proportional to the level of the corresponding target protein since it interacted with aptamer in the 1:1 ratio.

**Figure 6 biomedicines-10-00619-f006:**
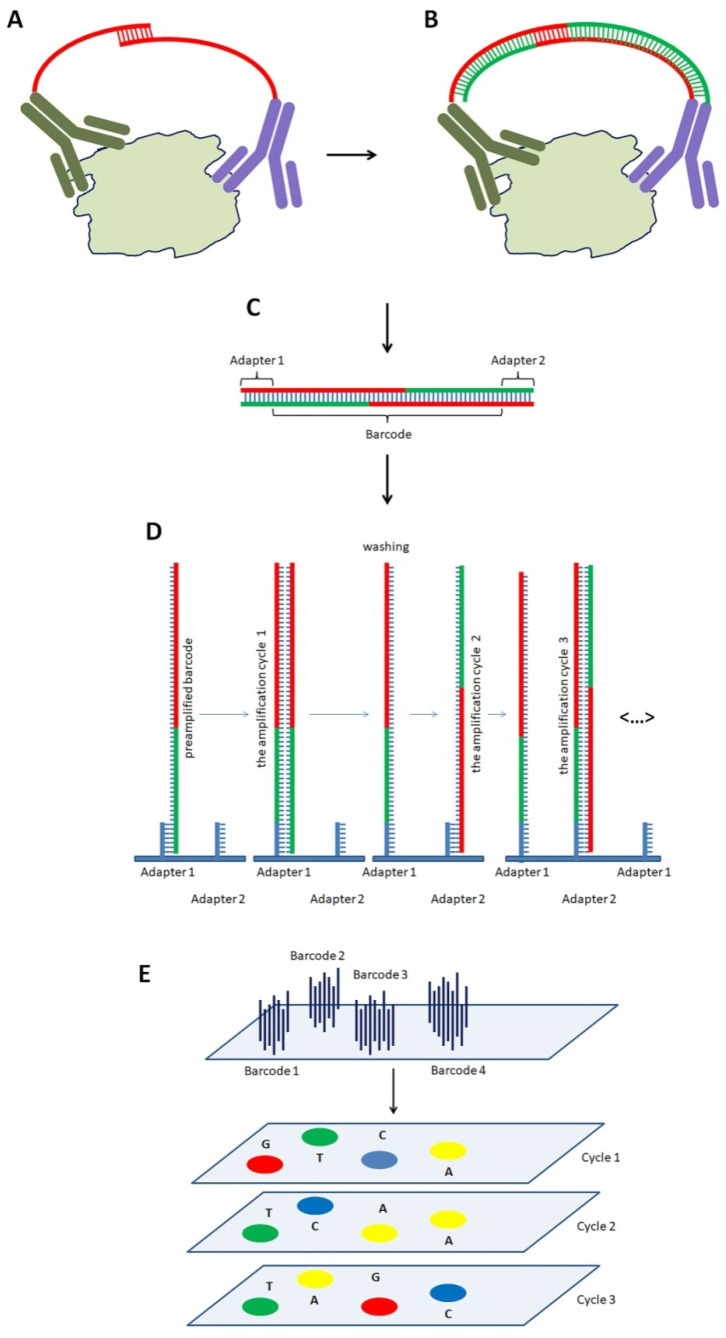
The Proximity Extension Assay. The experiment includes several steps. (**A**). Immunoassay (Step 1). The pairs of specific antibodies conjugated with ssDNA oligonucleotides specifically interact with target proteins. As both ssDNA contain short complementary sequences, they interact and form a duplex in the middle. (**B**). Extension of ssDNA (Step 2). The duplex serves as a set of primers to DNA polymerase that extends ssDNA to dsDNA. (**C**). A cleavage of dsDNA (Step 3). Cutting dsDNA from the antibodies produces an oligonucleotide. Each oligonucleotide contains a unique sequence that serves as a barcode to identify the target protein. It also has two short adapter sequences at the ends. (**D**). Preparation for the sequencing of the barcodes (Step 4). The obtained double-stranded oligonucleotides are denatured and interact with immobilized probes complementary to the adapters. Then, DNA polymerase amplifies DNA using the probes as primers. (**E**). Clusterization of DNA and sequencing the barcodes (Step 5). As the concentration of DNA is relatively low, the amplified molecules form clusters. Each cluster originates from a single DNA molecule and represents a specific barcode. As the “clusters” have to be homogenous on their composition, the dsDNA is denatured to wash out the disconnected ssDNA. In addition, DNA attached to the slide through one of the adapters (e.g., adapter 2) becomes cut and removed. The following sequencing of the DNA identifies and quantifies the barcodes. The numbers of identified barcodes are proportional to the levels of the corresponding target proteins since the single interaction of specific antibodies to a target protein produces only one barcode.

**Table 1 biomedicines-10-00619-t001:** Previously performed proteomic studies in psoriasis.

Year	Author/Reference	Proteomics Methodology	Samples	Patients	Key Findings
2005	Carlén et al. [10]	2D-electrophoresis, MALDI-TOF MS, Q-TOF-MS/MS	Skin	Psoriasis patients: lesional skin—7 samples; normal-looking skin—3 samples); patients with acute glutate psoriasis: lesional skin—6 samples; normal-looking skin—5 samples; healthy individuals—4 samples.	The first known proteomic study of lesional psoriatic skin; the first comparative analysis of patients with plaque psoriasis and acute guttate psoriasis.
2007	Bonnekoh et al. [11]	Multi-epitope ligand cartography (MELC) robot technology	Skin	Psoriasis patients: 6 samples of lesional skin; healthy volunteers 6 samples of healthy skin.	The authors showed a significant diversity in location of inflammatory epitopes after immunotherapy with efalizumab. They proposed CD138 and TRF1/CD71 as prognostic biomarkers of treatment outcome with efalizumab.
2007	Plavina et al. [12]	M-LAC coupled with LC-MS/MS	Serum	Psoriasis patients—20; healthy volunteers—20.	Depletion of immunoglobulins and albumin revealed upregulation of cytoskeletal and actin-binding proteins in plasma of psoriasis undetectable in regular serum.
2008	Plavina et al. [13]	M-LAC coupled with LC-MS/MS; ultracentrifugation followed by bynano LC-MS/MS	Serum	Psoriasis patients—20; healthy volunteers—20.	The authors identified 21 DEPs previously associated with other autoimmune disorders (e.g., thymosin β4, talin 1, γ-actin, filamin, profilin, S100A8, and S100A9) in serum of psoriasis patients. These DEPs were previously undetectable in the serum due to a higher abundance of immunoglobulins and serum albumin.
2011	Ryu et al. [14]	2D-electrophoresis, nanoLC-MS/MS	Two pools of skin samples (*n* = 8 and 28)	Psoriasis patients, 36 paired samples of lesional and normal-looking skin.	One of the first comparative analyses of lesional and normal-looking skin; the first known ontology analysis of DEPs in psoriatic skin.
2010	Piruzian et al. [15]	2D-electrophoresis, nanoLC-MS/MS	Pooled skin samples	Psoriasis patients, 3 paired samples of lesional and normal-looking skin.	The authors reported of the 10 most upregulated proteins in lesional skin.
2013	Schonthaler et al. [16]	iTRAQ-2DLC-MS/MS	Epidermis	Psoriasis patients, 19 paired samples of lesional and normal-looking skin.	The authors identified S100A8, S100A9, and complement C3 as the three most upregulated proteins in lesional skin: they also showed that knocking S100A9 out in *JunB*-*Jun* double knockout mice attenuated psoriasis-like skin disorder.
2013	van Swelm et al. [17]	MALDI-TOF MS	Urine	Psoriasis patients—60, with and without liver fibrosis	The authors proposed ITIH4 and CDH2 as candidate biomarkers of methotrexate-induced hepatic fibrosis in psoriasis patients.
2013	Williamson et al. [18]	Dimethyl labelling, LTQ-Orbitrapnano LC-MS/MS	Skin and serum	Psoriasis patients—4 paired samples of lesional and normal-looking skin; 4 samples of blood plasma; commercial blood plasma of healthy donors (*n* = 2).	The authors identified several dozen DEPs comparing lesional vs. normal-looking skin of psoriasis patients: they also proposed profilin as a biomarker of psoriasis.
2014	Fattahi et al. [19]	MALDI/TOF-TOF	Serum	Psoriasis patients—20; and 16 healthy volunteers.	The authors found a lower abundance of retinol-binding protein RBP4, a higher abundance of KRT10 and the unique expression pattern of α1 antitrypsin isoforms in sera of psoriasis patients.
2015	Cretu et al. [20]	LC-MS/MS	Pooled skin samples (*n* = 5)	Psoriasis patients with and without psoriatic arthritis (10 patients in each group), samples of lesional and normal-looking skin.	The authors proposed ITGB5 as a potential biomarker of psoriatic arthritis in psoriasis patients.
2015	Swindell et al. [21]	Label-free LC-MS/MS, LTQ-Orbitrap nanoLC-MS/MS	Skin	Psoriasis patients—14 paired samples of lesional and normal-looking skin.	The first known “bi-omic study” of psoriatic skin. The authors identified 748 DEPs in lesional and normal-looking psoriatic skin. They also discovered a modest correlation between protein and gene expression in psoriasis patients and characterized the role of IL-17A in disease-associated gene expression.
2016	Reindl et al. [22]	LTQ-Orbitrap nanoLC-MS/MS	Serum	Psoriasis patients—6; healthy volunteers—6.	The authors proposed AZGP1, complement C3, polymeric immunoglobulin receptor PIGR, and plasma kallikrein KLKB1 as disease-associated biomarkers. They discovered a moderate correlation between disease severity and the expression of DSP, complement C3, PIGR, and KRT17.
2017	Brunner [23]	Proximity extension assay	Serum	Patients with atopic dermatitis—59; psoriasis patients—22.	The authors found that inflammatory potential in patients with atopic dermatitis is higher than in psoriasis patients. They also showed a higher risk of cardiovascular disorders in both groups of patients.
2017	Kolbinger et al. [24]	Proximity extension assay	Serum, skin and dermis	Psoriasis patients—8 paired samples of lesional and normally-looking skin; healthy volunteers 8 skin samples; blood serum of the same individuals.	The authors showed how increased expression of antimicrobial peptides, proinflammatory cytokines and neutrophil chemoattractants normalizes in psoriasis patients after their treatment with secukinumab. They also proposed DEFB4 as a biomarker of the therapeutic response.
2017	Matsuura et al. [25]	MALDI-TOF MS, TripleTOF-MS/MS	Serum	Psoriasis patients with and without psoriatic arthritis (*n* = 10 and 24, respectively); 14 patients with atopic dermatitis; 23 healthy volunteers.	The authors identified several psoriasis/psoriatic arthritis associated DEPt originated from FGA, FLG, TMSB4X, and FLJ55606 in the sera of psoriasis patients.
2017	Méhul et al. [26]	qTOF-MS/MS and protein array	Stratum corneum	40 paired samples stripped from lesional and normal-looking skin of psoriasis patients.	The first comparative study of stratum corneum of psoriasis patients; the authors proposed 21 candidate biomarkers of lesional psoriatic stratum corneum.
2017	Méhul et al. [27]	qTOF-MS/MS and protein array	Stratum corneum	Patients with CTCL—10; psoriasis patients—24 (paired samples of stratum corneum stripped from lesional and normal-looking skin).	The first comparative proteomic study of patients with psoriasis and CTCL. The authors established a molecular signature of 112 DEPs to distinguish the samples of psoriasis patients and patients with CTCL.
2017	Wang et al., [28]	SomaScan	Serum	Patients with atopic dermatitis—20; patients with contact dermatitis—10; patients with atopic and contact dermatitis—10; psoriasis patients—12.	The authors reported 4 DEPs, namely KYNU, LG3BP, TPSB2, and CA6 associated with psoriasis and proposed KYNU as a disease-associated biomarker.
2018	Gęgotek et al. [29]	GeLC-MS/MS, LTQ Orbitrap-nanoLC-MS/MS	Serum	Psoriasis patients—6; healthy volunteers—6.	The authors detected a higher level of adducts in plasma of psoriasis patients. They also found a decreased level of vitamin D and proteins involved in lipid metabolism. In addition, they demonstrated higher abundance of proteins involved in immune response and signal transduction.
2018	Kim et al. [30]	Proximity extension assay	Serum	Psoriasis patients—266.	The authors discovered the strongest correlation between PASI and the expression levels of IL17A and IL17C, IL20, and CCL20 among the reponders to tofacitinib.
2018	Li et al. [31]	TMT labeling, LC-MS/MS	PBMC	New onset psoriasis patients (*n* = 31) and healthy volunteers (*n* = 32).	The authors identified new disease-associated proteins, namely ATM, SLFN5, ZNF512, SPATA13, DOCK2, ARSB, VIRMA, and NRGN.
2019	Foulkes et al. [32]	SomaScan	Serum	Psoriasis patients—10.	The authors reported increased expression of TNF- and interferon-dependent proteins.
2019	Garshick et al. [33]	Proximity extension assay	Serum and endothelial cells of brachial vein	Psoriasis patients—20.	The authors established a molecular signature of 8 DEPs, namely IL1β, CXCL10, VCAM-1, IL-8, CXCL1, LTB, ICAM-1, and CCL3, that characterizes the risk of atherosclerosis in psoriasis patients. They also discovered a correlation of the named biomarkers and PASI. In addition, they proposed an existence of a mechanism that damages different tissues in psoriasis.
2019	Gęgotek et al. [34]	GeLC-MS/MS, LTQ Orbitrap-nanoLC-MS/MS	Keratino-cytes and lympho-cytes	Psoriasis patients—6; healthy volunteers—6.	The authors discovered a higher level of adducts in plasma, skin and primary cells of psoriasis patients, lower expression of TXNRD1, higher expression of the glycolytic isoenzymes, namely PGAM1 and -2.
2019	Ge et al. [35]	UPLC-MS/MS with Q-Exactive Plus Hybrid Quadrupole-Orbitrap	Pooled skin samples (*n* = 15)	Psoriasis patients—45 paired samples of lesional and normal-looking skin.	The first study presenting a comprehensive analysis of 2-hydroxyisobutyrylation in lesional and normal-looking skin of psoriasis patients.
2019	Szél, E. et al. [36]	2D-electrophoresis, LC-MS/MS	Skin	Psoriasis patients—3 paired samples of lesional and normal-looking skin; healthy volunteers—3 skin samples,	The authors reported ~30 DEPs previously not associated with the disease. They also proposed PRKDC and MYBBP1A as potential key regulators of hyperproliferation and altered differentiation of skin cells, stress, and immune response in psoriasis,
2019	Xu et al. [37]	Custome-made array of specific antibodies directed to disease associated biomarkers, DIA-MS	Serum	Psoriasis patients—16; healthy volunteers—23,	The authors designed and tested a custom-made array of antibodies specific to 112 previously discovered disease-associated biomarkers. They also found a moderate correlation of PASI and the expression of PI3, CCL22, and IL12B. In addition, they proposed three predictive biomarkers, namely FCN2, MIF, and MMP1, to identify the responders to the traditional Chinese medicine YinXieLing.
2020	Medvedeva et al. [38]	Proximity extension assay	Serum	Psoriasis patients—150.	The authors proposed IL17A and KLK7 as biomarkers of disease severity and they also established the molecular signature of 4 DEPs, namely KLK7, PEDF, MDC, and ANGPTL4, to predict the outcome of the therapy to apremilast.
2020	Gęgotek et al. [39]	GeLC-MS/MS, LTQ Orbitrap-nanoLC-MS/MS	Fibroblasts	Psoriasis patients—5 samples of lesional skin; healthy volunteers—6 skin samples.	The authors discovered a higher abundance of TXNRD1 and a lower abundance of several glycolytic enzymes (PK, PGK2, ALDOL, and GAPDH) in dermal fibroblasts.
2020	Li et al. [40]	TMT labeling, LC-MS/MS	Skin	Psoriasis patients—11 samples of lesional skin; healthy volunteers—11 skin samples.	The authors identified 9 DEPs previously not associated with psoriasis: MPO, TYMP, IMPDH2, GSTM4, and ALDH3A1 that were upregulated and CES1, MAOB, MGST1, and GSTT1—that were downregulated in lesional skin.
2020	Zhou et al. [41]	iTRAQ-Labeling, LC-MS/MS	Skin and serum	Psoriasis patients—16 paired samples of lesional and normal-looking skin; 32 blood samples; healthy volunteers—15 skin samples and 24 blood samples.	The authors identified 4 new proteins, namely OAS2, IFIT3, IRF3, and MeCP2, previously not associated with psoriasis, and proposed OAS2 as a disease-associated biomarker to analyze both skin and sera samples. They also presented an optimized protocol for the obtaining of skin samples and their processing.
2021	Elnabawi et al. [42]	Proximity extension assay	Endothe-lial cells of brachial vein and serum	Psoriasis patients—23; healthy volunteers—10.	The authors found that the expression of CCL20 and IL6 correlates with LDL-cholesterol, endothelial inflammation score and PASI. They proposed CCL6 as a biomarker of impaired vascular health in psoriasis patients.
2021	Glickman et al. [43]	Proximity extension assay	Serum	Patients with moderate-to-severe alopecia areata (*n* = 35), atopic dermatitis (*n* = 49), moderate-to-severe psoriasis (*n* = 19) and healthy volunteers (*n* = 36).	The authors found that patients with the advanced forms of alopecia areata exhibit the highest systemic inflammatory tone and higher expression of cardiovascular risk biomarkers compared to the other groups of patients and their fellow groupmates without total involvement. They presented evidence that alopecia areata is a systemic disorder.
2021	Kaiser et al. [44]	Proximity extension assay	Serum	Psoriasis patients with and without the signs of atherosclerosis—85.	The authors discovered a negative correlation of GDF15 and vascular inflammation in the ascending aorta and entire aorta. They found that the expression of GDF15 positively correlates with carotid intima-media thickness and coronary artery calcium score in psoriasis patients without cardiovascular disease and statin treatment.
2021	Leijten et al. [45]	Proximity extension assay	Serum	Psoriasis patients with and without psoriatic arthritis (*n* = 20 and 18, respectively); healthy volunteers (*n* = 19).	The authors discovered strong correlations of joint swollenness and the expression levels of of ICAM-1 and CCL18. They also found a strong correlation of PASI and the expression of PI3 and IL17RA.
2021	Navrazhina et al. [46]	Proximity extension assay	Serum	Patients with moderate-to-severe hidradenitis suppurativa (*n* = 11), patients with psoriasis (*n* = 10) and healthy volunteers (*n* = 10).	The authors discovered that patients with hidradenitis suppurativa exhibited a significantly more intense inflammatory burden and an increase in cardiovascular/atherosclerosis-related biomarkers than psoriasis patients. They also proposed a computer model to distinguish sera samples of patients with hidradenitis suppurativa and psoriasis.
2021	Sobolev et al. [47]	LC-MS/MS	Skin	Psoriasis patients—5 paired samples of lesional and normal-looking skin; healthy volunteers—5 skin samples.	The authors proposed an existence of two adaptive mechanisms in normal-looking skin aimed to modulate there the development of the inflammatory response and accelerate the protein metabolism in the diseased cells, respectively. They reported a suppression of kallikrein-kinin system in normal-looking skin.
2021	Sobolev et al. [48]	LC-MS/MS	Skin	Psoriasis patients—5 paired samples of lesional and normal-looking skin; healthy volunteers—5 skin samples.	The authors discovered a set of 6 estrogen-dependent DEPs that modulate psoriasis in female skin. They proposed an existence of adaptive mechanism in female patients that facilitates the disease flow.
2021	Wang et al. [49]	TMT labeling, LC-MS/MS	pooled skin samples (*n* = 15)	Psoriasis patients—30 paired samples of lesional and normal-looking skin; healthy volunteers—30 skin samples.	The study compared psoriasis patients of Chinese and Caucasian descent pointing to the differences in protein expression in both populations. They identified GSTP1, SFN, KRT77, FLG2, and TREX2 as DEPs differentially expressed in Caucasians and SIRT1—as differentially expressed in Chinese patients.
2021	Zue et al. [50]	iTRAQ-Labeling, LC-MS/MS	PBMC	Two groups of 4 psoriasis patients with and without psoriatic arthritis.	The authors proposed SIRT2 as a potential biomarker of psoriatic arthritis in psoriasis patients.
2022	Navrazhina et al. [51]	Proximity extension assay	Skin	Patients with hidradenitis suppurativa—13 paired samples of lesional and normal-looking skin; psoriasis patients—11 paired samples of lesional and normal-looking skin; healthy individuals—11 skin samples.	The authors found that skin inflammation in the patients with hidradenitis suppurativa extends far beyond the skin lesions and sustains on a comparable level. They provided evidence that hidradenitis suppurativa is a systemic disorder.

## Data Availability

Not applicable.

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
