# Peer review of "Proteomic Studies of Psoriasis"

_biomedicines, 2022, doi:10.3390/biomedicines10030619_

Round 1

Reviewer 1 Report

This manuscript by Sobolev et al. reviews recent proteomic studies that have contributed to the identification of markers, diagnostic/therapeutic targets, and to the understanding of the molecular basis of pathogenesis for psoriasis. They introduce several different methodologies of proteomics used for DEPs analysis of skin as well as blood from subjects with psoriasis. I believe this manuscript will provide information that will be of great interest from researchers and clinicians in this field.

My suggests follows below,

Major points.

Including more detailed information whether control skin samples were nonlesional samples or normal samples from healthy control would be great.

Although well documented in the text, including tables that summarize and compare the methodologies and results, (for examples, how many DEPs, and the list of genes identified from each proteomic studies) from different proteomic approaches would help to have a clearer idea for readers. I believe including this information would make this manuscript more citable.

Minor points,

Please confirm whether authors mean to PRDX1 or PRDX2 in the following text (line 78-79) “In addition, they linked a higher expression of GSPT1 and PRDX1 to the pre- 78 vention of cells damages caused by reactive oxygen species (ROS) [13].”

Please check spellings, spacing etc (for example, Line 124, should there be a comma between “corneum and Mehul” ?)

Author Response

Dear reviewer,
On behalf of my colleagues, I would like to thank you. Your comments and suggestions helped us significantly improve our manuscript.
To comply with your suggestions, we introduced Table 1 with the requested information and corrected the text manuscript. Primarily, we paid attention to what kind of control the authors of cited papers used. At the same time, we could not completely satisfy your request to put the exact numbers of DEPs into the Table for several reasons. First, the authors of early proteomic studies usually do not mention precise numbers of DEPs due to technical limitations of the experimental design. Second, the analysis of skin samples presumes that we compare three different skin phenotypes, namely lesional and normally-looking skin of psoriasis patients and skin of healthy volunteers. Thus, it requires us to report the results of three comparisons instead of one. We have to admit that it is still possible technically. However, it will make Table 1 even longer than its current version since it contains multiple published proteomic studies. Third, the authors of cited proteomic studies often compared patients with several diagnoses. These additional data will make Table 1 hard to look through and interpret.
Instead, we would introduce the current version of Table 1 to provide information on the key findings of each cited study, the participants, and the proteomics methodology.
In response to the comment about "pyridoxines," we would like to say the following. We believe that the authors meant PRDX2 since they entitled their study "Proteomic analysis of psoriatic skin tissue for identification of differentially expressed proteins: up-regulation of GSTP1, SFN, and PRDX2 in psoriatic skin". On the other hand, we want to acknowledge that Ryu et al. published their paper in 2011. For the last ten years, the Uniprot database came through multiple transformations. The curators of the Uniprot database modified many records and removed some others. For this reason, we warned the readers to be cautious about using the original annotations provided by Ryu et al. in the cited paper (lines 78-83).
Regarding the phrase you quoted, we would like to acknowledge that we replaced PRDX1 with PRDX2 in the revised version of the manuscript. We also placed a comma between "corneum" and "Méhul."

Reviewer 2 Report

In this paper, the authors presented a review of proteomic studies regarding psoriatic research. The review is very comprehensive and I enjoyed reading it. Regarding the pathogenesis of the disease, it could be insightful if the authors would discuss the role of leukocyte-derived koebnerisin (S100A15) and psoriasin (S100A7) in the inflammation of psoriasis (DOI: 10.1016/j.jdermsci.2015.05.007). 

Author Response

Dear reviewer,
On behalf of my colleagues, I would like to thank you. Your comments and suggestions helped us significantly improve our manuscript.
Regarding your comment, we would like to mention that we briefly discussed the role of leukocyte-derived koebnerisin and psoriasin in the inflammation of psoriasis. Please, see the revised manuscript (lines 455-464).